# Effects of Variation in Urine Sample Storage Conditions on 16S Urogenital Microbiome Analyses

Tanya Kumar,[a] MacKenzie Bryant,[b] Kalen Cantrell,[c,d] Se Jin Song,[d] Daniel McDonald,[b] Helena M. Tubb,[b] Sawyer Farmer,[b] Emily S. Lukacz,[e] Linda Brubaker,[e] Rob Knight[b,c,d,f]

[a]Medical Scientist Training Program, University of California San Diego, La Jolla, California, USA
[b]Department of Pediatrics, University of California San Diego, La Jolla, California, USA
[c]Department of Computer Science and Engineering, University of California San Diego, La Jolla, California, USA
[d]Center for Microbiome Innovation, Jacobs School of Engineering, University of California San Diego, La Jolla, California, USA
[e]Department of Obstetrics, Gynecology and Reproductive Sciences, University of California San Diego, La Jolla, California, USA
[f]Department of Bioengineering, University of California San Diego, La Jolla, California, USA

Tanya Kumar, MacKenzie Bryant, and Kalen Cantrell contributed equally to this article. Author order was determined based on each author's main focus of writing and the general order of the paper body.

**ABSTRACT** Replicability is a well-established challenge in microbiome research with a variety of contributing factors at all stages, from sample collection to code execution. Here, we focus on voided urine sample storage conditions for urogenital microbiome analysis. Using urine samples collected from 10 adult females, we investigated the microbiome preservation efficacy of AssayAssure Genelock (Genelock), compared with no preservative, under different temperature conditions. We varied temperature over 48 h in order to examine the impact of conditions samples may experience with home voided urine collection and shipping to a central biorepository. The following common lab and shipping conditions were investigated: −20℃, ambient temperature, 4℃, freeze-thaw cycle, and heat cycle. At 48 h, all samples were stored at −80℃ until processing. After generating 16S rRNA gene amplicon sequencing data using the highly sensitive KatharoSeq protocol, we observed individual variation in both alpha and beta diversity metrics below interhuman differences, corroborating reports of individual microbiome variability in other specimen types. While there was no significant difference in beta diversity when comparing Genelock versus no preservative, we did observe a higher concordance with Genelock samples shipped at colder temperatures (–20℃ and 4℃) when compared with the samples shipped at −20℃ without preservative. Our results indicate that Genelock does not introduce a significant amount of microbial bias when used on a range of temperatures and is most effective at colder temperatures.

**IMPORTANCE** The urogenital microbiome is an understudied yet important human microbiome niche. Research has been stimulated by the relatively recent discovery that urine is not sterile; urinary tract microbes have been linked to health problems, including urinary infections, incontinence, and cancer. The quality of life and economic impact of UTIs and urgency incontinence alone are enormous, with $3.5 billion and $82.6 billion, respectively, spent in the United States. annually. Given the low biomass of urine, novelty of the field, and limited reproducibility evidence, it is critical to study urine sample storage conditions to optimize scientific rigor. Efficient and reliable preservation methods inform methods for home self-sample collection and shipping, increasing the potential use in larger-scale studies. Here, we examined both buffer and temperature variation effects on 16S rRNA gene amplicon sequencing results from urogenital samples, providing data on the consequences of common storage methods on urogenital microbiome results.

**KEYWORDS** 16S, microbiome, sample storage, urobiome, urogenital microbiome

Address correspondence to Rob Knight, robknight@ucsd.edu.

The authors declare a conflict of interest. Emily S. Lukacz: Pathnostics

The discovery that the bladder is not sterile (1–4) has stimulated key advances in the urogenital microbiome field, which still remains a relatively understudied component of the human microbiome. The urogenital microbiome, assessed using voided urine, include bladder, urinary tract, and potentially genital microbes (5). Additional research is warranted as the urogenital microbiome has biologic plausibility in urinary tract health and disease; it has been tied to some of the most common urinary conditions such as lower urinary tract symptoms (LUTS), which include urinary incontinence and urinary tract infections (UTIs) (4, 6, 7). UTIs impact over 150 million people per year globally with an increasing prevalence of antibiotic resistance, recurrence, and associated serious health complications (8). Thus, an in-depth understanding of the urogenital microbiome through large-scale studies is necessary to better understand relationships between urinary health, LUTS prevention, and intervention strategies.

There is scant evidence informing optimal storage conditions of urinary samples. However, other niches, such as the fecal microbiome, have well-recognized replicability issues related to storage conditions. Investigation of the effects of various storage conditions on urine samples (9, 10) will be foundational for population-level studies relating the urogenital microbiome to bladder health. In this study, we evaluated the efficacy of AssayAssure Genelock (Genelock), a nucleic acid preservative formulated by Sierra Molecular currently used for clinical urogenital sample collection (11–13). Previously, Jung et al. (14) found temperature-dependent biases when examining Genelock on urine samples at −20°C, 4°C, and ambient temperature (~23°C). Also informed by the work of Song et al. (15) and Marotz et al. (16), we wanted to further investigate the impact of several temperature conditions on the urogenital microbiome when using Genelock as a preservative: −20°C, 4°C, ambient temperature, heat cycles, and freeze-thaw cycles (Text S1A; Fig. 1A).

Volunteers gave verbal consent under an IRB-approved study protocol (UCSD protocol no. 801735) after being provided with and reading through a written document. Ten healthy adult females donated a single, voided, urine sample, and aliquots were immediately transferred into tubes without Genelock as well as with Genelock, in a 1:10 volume of Genelock to urine. Samples were stored in their respective temperature condition and transported to the lab for further storage and processing (Text S1A; Fig. 1A). Urine samples and serially diluted positive KatharoSeq controls (Text S1B) were plated and extracted using Earth Microbiome Project standard protocols (https://earthmicrobiome.org/protocols-and-standards/), further outlined in Shaffer et al. (17) (Text S1C). Amplification of the 16S rRNA V4 region was performed using a miniaturized PCR protocol (18) then sequenced on an Illumina MiSeq (Text S1D). Forward read sequences were trimmed, filtered, and demultiplexed using Qiita (19) (Text S1E). The highly sensitive KatharoSeq protocol implements serially diluted bacterial mock community controls to utilize known read counts as a sample exclusion threshold, or limit of detection, allowing us to account for sequencing difficulty often encountered with low biomass samples, such as urine, due to potential trace contamination from DNA extraction or PCR kit reagents (20). We utilized the 50% KatharoSeq threshold to exclude and rarefy samples to 986 reads, resulting in a final analysis pool of 9 participants and 161 samples.

We first examined the beta diversity metrics of the samples (Fig. 1B). In both weighted and unweighted UniFrac, permutational multivariate analysis of variance (PERMANOVA) revealed that beta diversity was driven primarily by participant (PERMANOVA, unweighted $P = 0.001$, f = 17.85; weighted $P = 0.001$, f = 54.5), rather than preservative method (PERMANOVA, unweighted $P = 0.96$, f = 0.46; weighted $P = 0.43$, f = 0.87) or temperature treatment (PERMANOVA, unweighted $P = 0.76$, f = 0.84; weighted $P = 0.25$, f = 1.22). This result is consistent with previous reports on other microbiome sites (5, 9), which indicate that an individual's microbiome composition accounts for a large portion of beta diversity variation. We also

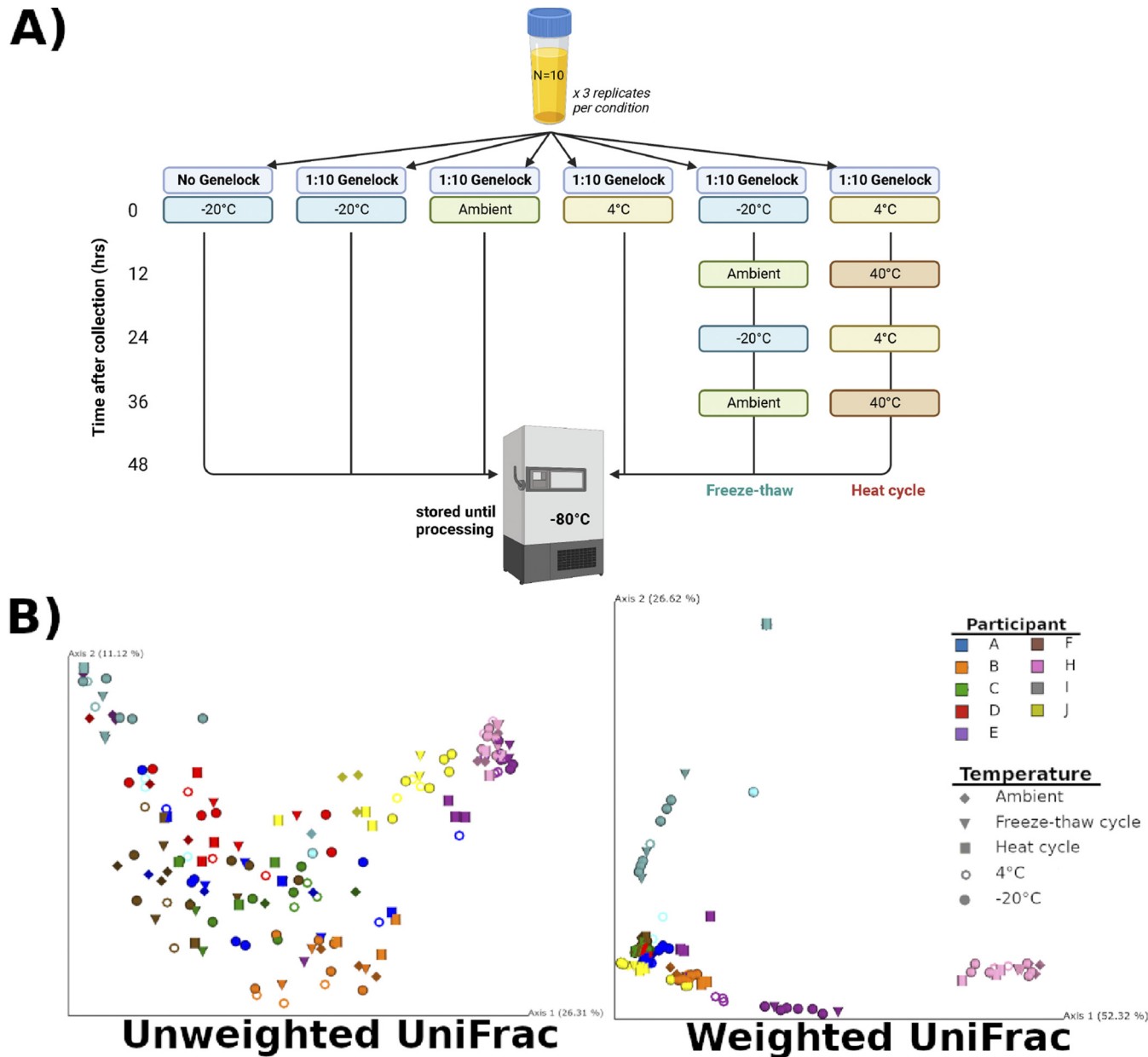

FIG 1 Experimental design and clustering by individual. (A) 10 healthy adult females gave a single 10-mL urine sample, which was processed in triplicate and subject to varying temperature conditions with the presence or absence of AssayAssure Genelock (Genelock). This figure was created with BioRender.com. (B) Principal-coordinate analysis plots of weighted and unweighted UniFrac distances.

observed that weighted UniFrac showed a higher degree of clustering than unweighted, implying that similarities in common rather than rare taxa are driving the clustering.

Next, we examined the effect of temperature by comparing the UniFrac distances between the different temperature treatment groups and samples stored immediately at −20°C without Genelock. Long-term urine storage is effective at −20°C (21) and is the sample collection procedure used in the lab (22), prompting us to use this as the comparator group. We observed the mean distance within each storage condition group is at or below the mean distance within each individual (interindividual) and below the mean distance between participants (interhuman), suggesting that the individual is the primary driver of beta diversity. Additionally, in the case of unweighted UniFrac and Jaccard, the distance is close

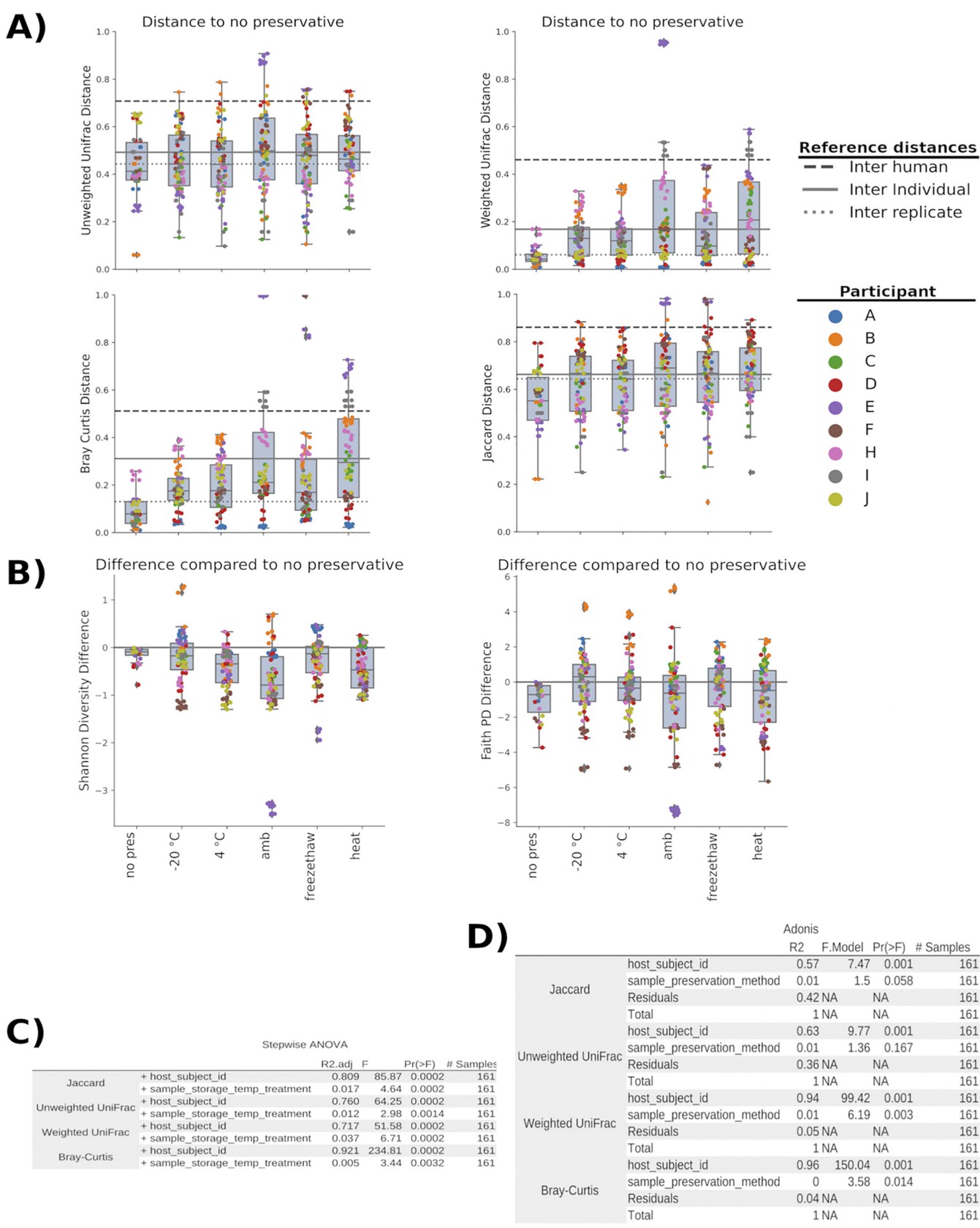

**FIG 2** Quantification of results. Effect of temperature treatment and presence of Genelock on urogenital microbiome composition. Genelock is present in all samples other than the comparator group, which was stored at −20°C with no preservative. (A) Distances between different temperature treatments and comparator group unweighted and weighted UniFrac, BrayCurtis, and Jaccard distances. Interhuman distance was calculated by

to the mean distance between replicates, suggesting temperature plays a minor role in the microbial composition (Fig. 2A).

The mean differences in both alpha diversity measures (Shannon and Faith PD) were close to 0 (Fig. 2B). This suggests that the individual is associated with major differences in both microbial community composition (Fig. 2A) and diversity (Fig. 2B). Fig. 2B also depicts individual diversity variations, with some participants (such as participant B) having higher deviations from the comparator group, while others (such as participant F) having an overall lower reading on richness, evenness, and phylogenetic-based diversity. Permutation of variance (stepwise ANOVA) was used to quantify the effects of individual variation in microbial composition (Fig. 2C). In all tests, individual variation accounted for the majority of variation (Jaccard [$R^2 = 0.81$], unweighted UniFrac [$R^2 = 0.76$], weighted UniFrac [$R^2 = 0.72$], and BrayCurtis [$R^2 = 0.92$]). This suggests that the host accounts for most of the microbial community composition. Additionally, the effects that temperature has on microbial composition were also tested and were shown to be small but statistically significant (Jaccard [$R^2 = 0.017$], unweighted UniFrac [$R^2 = 0.012$], weighted UniFrac [$R^2 = 0.037$], and Bray-Curtis [$R^2 = 0.05$]).

Finally, we looked at the effect of Genelock by comparing samples without Genelock (stored at $-20°C$) and samples with Genelock stored at $-20°C$ (Fig. 2D). ADONIS, a permutational multivariate analysis of variance test, showed there was no significant difference in preservation methods with unweighted UniFrac and Jaccard distances. Genelock has a small, but significant effect in both weighted UniFrac ($R^2 = 0.01$) and Bray-Curtis ($R^2 = 0.003$). In all metrics, the individual accounted for the most variance (Jaccard [$R^2 = 0.57$], unweighted UniFrac [$R^2 = 0.63$], weighted UniFrac [$R^2 = 0.94$], and Bray-Curtis [$R^2 = 0.96$]).

Overall, our results suggest that Genelock is a reasonable urine sample preservative for urogenital microbiome studies as it does not introduce more microbial bias than differences attributable to individual variation. Additionally, temperature extremes do not appear to impact diversity meaningfully when urine is preserved in Genelock for up to 48 h. These data should be used to inform the design of population-based studies.

Additionally, our findings support the growing observation that the urogenital microbiome may be a marker of interindividual microbial diversity. Future large-scale studies are warranted to understand individual urogenital microbiomes and their relationship to urinary tract health. Self-collection of voided urine samples for microbiome studies is feasible, facilitating study of large, nonclinical populations in order to advance our understanding of the urobiome.

The study has several limitations. First, the small sample size limits broad conclusions and is intended to inform larger population-level studies. Second, a limited number of conditions were studied. Thus, caution in extrapolating to other collection methods, shipping or storage conditions is warranted.

**Data availability.** Data is available in Qiita (19) under study ID 14383 and through the European Nucleotide Archive under study ID PRJEB53631. A STORMS (Strengthening The Organizing and Reporting of Microbiome Studies) 190 checklist (23) is available at 10.5281/zenodo.6788075.

## SUPPLEMENTAL MATERIAL

Supplemental material is available online only.
**TEXT S1**, DOCX file, 0.02 MB.

## FIG 2 Legend (Continued)

calculating the distance between each individual's samples to the other individual's samples (distances between samples from the same individual were not taken into account) and taking the median distance. Interreplicate distance was calculated by finding the distance between replicates (replicates are samples from the same individual that were treated with the same preservative method and stored at the same temperature) and taking the median replicate distance. (B) Shannon and faith PD alpha diversity differences between different temperature treatments and the comparator group. (C) Values are based on permutation tests of variance (stepwise ANOVA). (D) ADONIS (a multivariate analysis of variance) test. For this test Genelock samples stored at $-20°C$ were compared against the comparator group samples ($-20°C$; no preservative) to capture the effect of the individual and preservative.

## ACKNOWLEDGMENTS

We thank Karenina Sanders for help in formatting citations and paper submission, and Gail Ackermann for education and help in initial Qiita data processing.

This work was supported by the following grant: NIH T32 GM719876.

E.S.L. is a consultant and advisory board member for molecular diagnostics laboratory company Pathnostics. All other authors declare no conflicts of interest.

T.K., data analysis, manuscript writing, figure production; M.B., study coordination, manuscript writing; K.C., data analysis, manuscript writing, figure production; S.J.S., data analysis; D.M., data analysis; H.M.T., sample collection; S.F., sample processing; E.S.L., study coordination and oversight, manuscript writing; L.B., study coordination and oversight, manuscript writing; R.K., study oversight and manuscript compilation.

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
