## [Reviewer comments · mSystems]

Effects of Variation in Urine Sample Storage Conditions on 16S Urogenital Microbiome Analyses

Tanya Kumar, MacKenzie Bryant, Kalen Cantrell, Se Jin Song, Daniel McDonald, Helena Tubb, Sawyer Farmer, Emily Lukacz, Linda Brubaker, and Rob Knight

Corresponding Author(s): Rob Knight, University of California, San Diego

Review Timeline:

Submission Date:	October 24, 2022
Editorial Decision:	November 1, 2022
Revision Received:	November 4, 2022
Accepted:	November 7, 2022

Editor: Jotham Suez

Reviewer(s): The reviewers have opted to remain anonymous.

Transaction Report:

DOI: <https://doi.org/10.1128/msystems.01029-22>

October 31, 2022

Dr. Rob Knight
University of California, San Diego
Pediatrics
La Jolla, CA

Re: mSystems01029-22 (Effects of Variation in Urine Sample Storage Conditions on 16S Urogenital Microbiome Analyses)

Dear Dr. Rob Knight:

Thank you for submitting your revised manuscript to mSystems. We have completed our review, and I am pleased to inform you that, in principle, we expect to accept it for publication in mSystems. I only have a few minor requests that should be addressed before final acceptance:

1. Text S1, line 10: Please replace the term "biological women". If the sex of the participants is the sole critical information, and all were assigned female at birth, please simply use "females". If it is critical to include gender information, please use "cisgender women". Please make sure the main text agrees with the supplement (currently the main text only says "women").
2. Since the text mentions that the participants were healthy, it would be helpful to add to Text S1 the inclusion and exclusion criteria.
3. Based on reviewers' comments, "AssayAssure" was replaced throughout the text with "Genelock", however, figure 1A still mentions "AA" - please replace with "Genelock".

Thank you for the privilege of reviewing your work. Below you will find instructions from the mSystems editorial office.

Preparing Revision Guidelines

- Each figure must be uploaded as a separate file, and any multipanel figures must be assembled into one file.
- Manuscript: A .DOC version of the revised manuscript
- Figures: Editable, high-resolution, individual figure files are required at revision, TIFF or EPS files are preferred

Sincerely,

Jotham Suez

Editor, mSystems

Journals Department
We thank you for your incredibly helpful comments, which have strengthened our manuscript. Please see our point-by-point responses to the suggestions we have received. Comments provided given in plain text (black), while author responses are in red.

Points to be Addressed (Comments for the Author):

1. Text S1, line 10: Please replace the term "biological women". If the sex of the participants is the sole critical information, and all were assigned female at birth, please simply use "females". If it is critical to include gender information, please use "cisgender women". Please make sure the main text agrees with the supplement (currently the main text only says "women").

Thank you for the suggestion – We have changed “biological women” to “females” in Text S1 (L13), the main manuscript (L38 and L94), and the legend of Figure 1A (L197).

2. Since the text mentions that the participants were healthy, it would be helpful to add to Text S1 the inclusion and exclusion criteria.

We have added the following inclusion criteria in Text S1 (L4-L7): “Our inclusion criteria for “healthy” was defined by the participant being free of symptoms of disease such as fever or chills, cough, shortness of breath, headache, loss of taste or smell, sore throat, congestion, nausea or vomiting, and diarrhea.”

3. Based on reviewers' comments, "AssayAssure" was replaced throughout the text with "Genelock", however, figure 1A still mentions "AA" - please replace with "Genelock".

Thank you for pointing this out. “AA” in Figure 1A has been replaced with Genelock. Additionally, the figure legend has been updated in the main manuscript file (L199).

November 7, 2022

Dr. Rob Knight
University of California, San Diego
Pediatrics
La Jolla, CA

Re: mSystems01029-22R1 (Effects of Variation in Urine Sample Storage Conditions on 16S Urogenital Microbiome Analyses)

Dear Dr. Rob Knight:

Your manuscript has been accepted, and I am forwarding it to the ASM Journals Department for publication. For your reference, ASM Journals' address is given below. Before it can be scheduled for publication, your manuscript will be checked by the mSystems production staff to make sure that all elements meet the technical requirements for publication. They will contact you if anything needs to be revised before copyediting and production can begin. Otherwise, you will be notified when your proofs are ready to be viewed.

Publication Fees:

If you would like to submit a potential Featured Image, please email a file and a short legend to mSystems@asmusa.org. Please note that we can only consider images that (i) the authors created or own and (ii) have not been previously published. By submitting, you agree that the image can be used under the same terms as the published article. File requirements: square dimensions (4" x 4"), 300 dpi resolution, RGB colorspace, TIF file format.

We recognize that the video files can become quite large, and so to avoid quality loss ASM suggests sending the video file via <https://www.wetransfer.com/>. When you have a final version of the video and the still ready to share, please send it to mSystems staff at mSystems@asmusa.org.

Sincerely,

Jotham Suez

Editor, mSystems

Journals Department
Text S1: Accept